# Shallow Learning In Materio.

## Abstract

We introduce Shallow Learning In Materio (SLIM) as a resource-efficient method to realize closed-loop higher-order perceptrons. Our SLIM method provides a rebuttal to the Minsky school's disputes with the Rosenblatt school about the efficacy of learning representations in shallow perceptrons. As a proof-of-concept, here we devise a physically-scalable realization of the parity function. Our findings are relevant to artificial intelligence engineers, as well as neuroscientists and biologists.

## 1 Introduction

How do we best learn representations? We do not yet fully understand how cognition is manifested in any brain, not even in those of a worm (Rankin, 2004). It is an open question if the shallow brain of a worm is capable of working memory, but if it were then it certainly must depart from the mechanistic models of large-scale brains (Eliasmith et al., 2012). Nevertheless, worm-brain inspired learning combined with "scalable" deep learning architectures have been employed in self-driving cars (Lechner et al., 2020). At present, by scalable we refer to TPU-based architectures (Jouppi et al., 2017) trained by gradient-descent (Rumelhart et al., 1986). However, one could envision a super-scalable future that is less synthetic and based on self-organized nanomaterial systems (Bose et al., 2015; Chen et al., 2020; Mirigliano et al., 2021) that natively realize higher-order (Lawrence, 2022a) and recurrent neural networks. In this short communication, we shall lay yet another brick towards such a future by providing theoretical arguments.

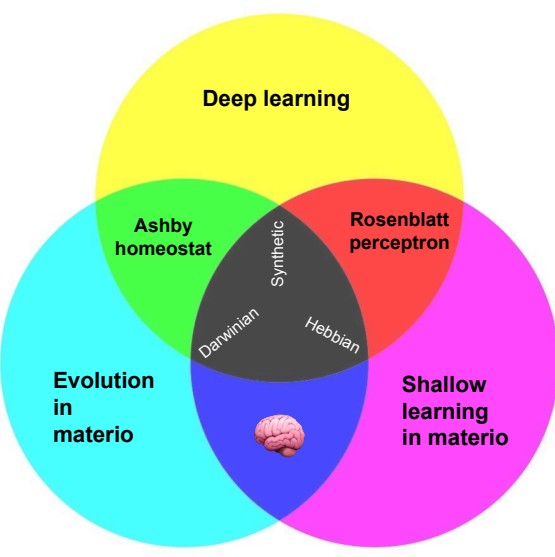

Figure 1: A typology of cognitive material systems.

Our perspective on cognitive material systems is illuminated in Figure 1. Deep learning owes its success to our technological capacity to synthesize massively-parallel and programmable electronic circuits. It is yet to fully exploit Darwinian and Hebbian learning methods that pioneers of the cybernetics movement experimented with by training homeostats (Ashby, 1952) and perceptrons (Rosenblatt, 1961). The spirit of Darwinian (Stanley et al., 2019) and Hebbian (Scellier & Bengio, 2017) learning continues to be alive, though. Here, we add fuel to that fire by advocating for an in-materio approach.

Employing physical systems in their native form for solving computational tasks had gained attention due to the efforts of the 'evolution in materio' community (Miller & Downing, 2002). The earliest result was by Pask (1960) who grew dendritic metallic threads in a ferrous sulphate solution to function as a sound-frequency discriminator (which he called an ear, quite romantically). Now, more recent efforts are under the banner of physical reservoir computing (Tanaka et al., 2019) for realizing sequential functionality. Here, we will commit to combinational functionality by equilibrium-point logic (Lawrence, 2022b) in material systems realizing closed-loop higher-order perceptrons.

## 2 Theory

Perceptrons were developed by Rosenblatt and his team, and were trained by a Hebbian learning rule (error-controlled reinforcement) with proven guarantees for convergence. Unfortunately, they started recieving a bad rap after Minsky & Papert (1988) published a proof that $2^N$ association neurons are required to learn the $N$-bit parity function. However, this analysis is only applicable if all neurons are threshold logic gates, what Rosenblatt called simple units. Physical neural networks, on the other hand, can natively realize complex units. Hence, we introduce a shallow learning in materio (SLIM) perceptron as depicted in Figure 2.

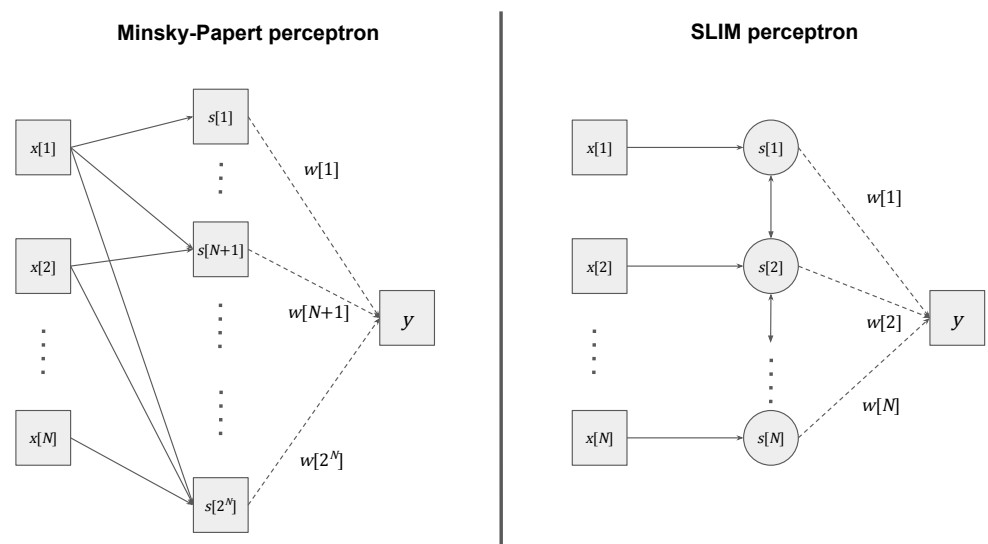

Figure 2: To learn the $N$-bit parity function, the required number of synaptic weights (depicted as dashed lines) for our SLIM perceptron scales as $N$ instead of the $2^N$ required for the Minsky-Papert perceptron. This gain in resource-efficiency is possible because the hidden states $s_{1:N}$ of the SLIM perceptron can compute deep-feedforward functionality by equilibrium-point control.

For a proof-of-concept, we commit to a minimally connected recurrent network with physical states $s_i$ from $i = 1 : N$, yielding a state-space model of the form

$$\dot{s}_i = x_i + F_i(s_{i-1}, s_i, s_{i+1}), \tag{1}$$

where $F_i$ is a nonlinear function. We conjecture that all possible $N$-bit functions may be realized if arbitrary choices of $F_{1:N}$ are allowed. At present there is no engineering theory to design an optimal $F_i$ (even when $N = 2$).

We first take an approach amicable to discrete mathematics, and demonstrate equilibrium-point logic in Figure 3 with $F_{1:2}$ designed as piecewise-constant functions. A promising approach to obtain $F_i$ for higher dimensions is to identify an analogy with cellular automaton that are capable of equilibrium-point parity logic in arbitrary dimensions (Betel et al., 2013). Obtaining scaling laws for the volume of state-space in action during equilibrium-point logic may be another worthy problem to ponder upon.

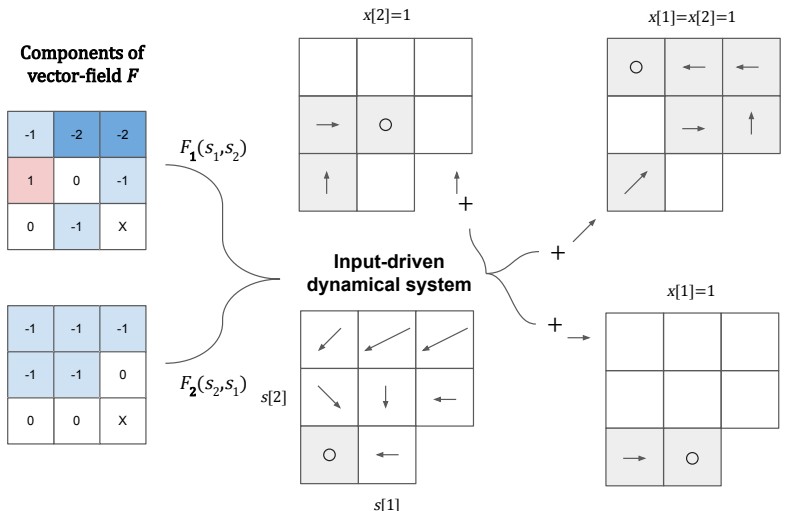

Figure 3: Equilibrium-point logic in a piecewise-constant vector field $\dot{s}_i = x_i + F_i(s_1, s_2)$ designed to realize $s_1 = x_1 \oplus x_2$. The components of $F_i$ combine to generate the field at $x_1 = x_2 = 0$ for the input-driven dynamical system. The field can be updated on addition of input(s) in 3 possible directions, and the system traverses to a new equilibrium-point as shown (traversing the gray regions).

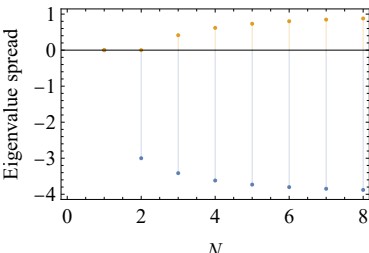

Figure 4: Maximum and minimum eigenvalues over all $2^N$ locally-linear modes of equation 2.

Imposing conditions of physical realizability on $F_i$ would affect the neuronal capacity (Baldi & Vershynin, 2018) of our SLIM perceptron. To obtain an insight into the abundance of unique functions expressable by SLIM, let us consider a unit-resistor learnable-threshold ($w_i$) diode network of the form

$$F_i(s_{i-1}, s_i, s_{i+1}) = s_{i-1} + s_{i+1} - 2s_i - \text{Ramp}(s_i - w_i), \quad (2)$$

with $s_0 \equiv s_1$ and $s_{N+1} \equiv s_N$. The above equation is the simplest expression that captures the nonlinear synergetic interactivity found in the Lyapunov-stable resistor tunnel-diode networks studied in

(Lawrence, 2022b). For each $i = 1 : N$, depending on $\operatorname{sgn} s_i - w_i$, there is a positive or a negative mode of equation 2, and thus there are $2^N$ modes of convergence to equilibrium. Each mode has $N$ eigenvalues, thus there are $N2^N$ different timescales. The smallest and largest eigenvalues are plotted in Figure 4, and the eigenvalue spread is larger for higher $N$. Because the largest eigenvalue is positive, while the system is Lyapunov stable, we may expect a non-trivial mixing of the modes of functionality on the way to equilibrium. This was confirmed empirically for $N = 8$ and 1000 random arrays of weights with $w_i \in (0, 1)$. No two weight arrays yielded the same mode of equilibriation for all $2^N = 256$ inputs, and thus 1000/1000 functions expressed were unique (for $N = 3$ this was not true and only 266/1000 unique functions were expressed). Wolfram Mathematica code to reproduce this result and investigate it for other values of $N$ is provided in the Appendix.

# 3 Conclusion

Our contribution here is threefold: (1) a typology of cognitive material systems that puts a spotlight on yet-to-be-appreciated alternatives to deep learning, (2) a mathematically tractable framework to investigate recurrent networks for deep feedforward functionality, (3) framing open problems in equilibrium-point logic. More theory is needed to develop constructive high-dimensional examples, and a statistical analysis of SLIM's performance. Next steps could be to obtain estimates on the learning duration, and check if it is superior to estimates obtained from the principal convergence theorem for perceptrons (Rosenblatt, 1961, Section 5.5, Theorem 4). Given the well established (in silico) deep learning industry, even with a more practical demonstration, business economics would prevent the shift to a SLIM paradigm in the near future. Nevertheless, the SLIM concept may act as a catalyst for gifted mathematicians to make new connections or help neuroscientists in unravelling the mysteries of small-scale brains.

# Reviewer contributions

Reviewer *sByK* asks why this work should be considered novel, in comparison to earlier concepts such as predictive coding networks. The novelty here comes from using the function $F_i$ in equation 1 to efficiently realize nonlinear predictors in materio, an improvement over the linear weighted-sum predictors (Srinivasan et al., 1982) that were inspired from image-compression techniques designed for conventional computers.

Reviewer *joB1* is kind to provide a thoughtful summary, and suggests to compare this work to two other alternatives for realizing the parity function : the complex-weighted neuron of Aizenberg (2008) and the translated multiplicative neuron of Iyoda et al. (2003). In both alternatives, the implementation would be less robust to noise at large $N$, because only a single neuron is employed (a robust implementation would require a circuit of many physical units for the neuron, making it "single" only in a mathematical sense). SLIM need not suffer from such crowding problems, because the state-space can grow exponentially in volume with $N$. Several provably convergent schemes of Hebbian learning as given by Pineda (1987) may be engineered in materio, to act as a generalization of backpropagation for closed-loop higher-order perceptrons.

Based on feedback from all reviewers, the technical novelty of this work has been clarified in the conclusion (contribution no. 2).

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

# A Appendix

Snippet of Wolfram Mathematica code used to estimate the abundance of in-materio functionality.

```
(*Define the material's function by starting from the all-0 state and updating over 100 loops*)
F[s_] := (t = ArrayPad[s, 1, "Fixed"]; 2. * s - t[[3 ;; ]] - t[[ ;; -3]]);
settleMaterio[n_, g_, x_] := Nest[# + .01 (x - F[#] - Ramp[# - g]) &, 0 * x, 100];
funMaterio[n_, g_] := FromDigits[Flatten@Table[UnitStep[settleMaterio[n, g, x] - g], {x, Tuples[{0, 1}, n]}], 2];

(*Count the number of distinct functions realised over 1000 random biases*)
abundance[n_] := CountDistinct@ParallelTable[funMaterio[n, g], {g, RandomReal[1., {1000, n}]}];

abundance[3]
266

abundance[8]
1000
```

