# OpenReview forum: "Shallow Learning In Materio."
_ICLR.cc/2023/Conference — Submitted to ICLR 2023_

### Official Review · Reviewer_joB1 · 2022-10-14

**Confidence:** 3
**Correctness:** 2
**Technical Novelty And Significance:** 2
**Empirical Novelty And Significance:** 2
**Recommendation:** 3

**Clarity, Quality, Novelty And Reproducibility:**

Originality:
* The main idea of the proposed approach - to apply the recurrent shallow neural networks in the context of in-materio technologies (enabling to use higher-order units) - is novel and interesting.
However, the realization is not very convincing (due to the lack of clear description, deeper theoretical or experimental evaluation or comparison to alternative approaches).

Related works seem to be cited adequately, except:
- Based on Section 1, the paper closely follows (Lawrence, 2022b). However, (Lawrence, 2022b) seems to be unavailable (the link doesn't work and I was not able to find it elsewhere)
 - Because the authors concentrate on the n-bit parity function realized by shallow networks, they should compare their approach to some alternative/similar models and cite them (e.g., [1], [2]).



Clarity and reproducibility:
* Both the proposed model and the experiment should be described in more detail.
There are many open questions that hinder reproducibility (e.g., missing description of the training process, missing detailed description of the used F-functions,...).
* Figure 3 is not well-described and thus harly comprehensible.

Quality:
* The technical quality of the paper is poor. The paper looks like an incomplete work in progress (see "Strength And Weaknesses" Section for details).

Minor:
* (Rumelhart et al., 1986) is cited as a reference to TPU-based architectures, isn't that an error?

[1] Aizenberg, I. Solving the XOR and parity N problems using a single universal binary neuron. Soft Comput 12, 215–222 (2008). https://doi.org/10.1007/s00500-007-0204-9

[2] Iyoda, E.M., Nobuhara, H. & Hirota, K. A Solution for the N-bit Parity Problem Using a Single Translated Multiplicative Neuron. Neural Processing Letters 18, 233–238 (2003). https://doi.org/10.1023/B:NEPL.0000011147.74207.8c




**Strength And Weaknesses:**

Strengths:
1. I agree with the authors that the shallow recurrent higher-order neural networks are a promising area of future research - due to the progress of in-materio technologies.
2. The illustrative example is well-chosen. It demonstrates well how the proposed SLIM model could overcame the classical problem of n-bit parity (Minsky&Pappert,1988) - thanks to the higher-order units used instead of threshold logic gates.
3. The authors provide several open questions concerning their model as possible subjects of future research.

Limitations:
1. The paper is very brief  and vague in every aspect and its techniqual quality is imited.
2. There is no theoretical justification of the method (and its convergence) and the empirical evaluation is limited to a toy problem.

Discussion of the limitations:
1. The paper is very brief  and vague in every aspect and its techniqual quality is imited.
  * Mainly, the proposed model should be described in more detail, including training algorithm and analysis/discussion of the choice of the F_i functions.
  * The illustrative example also needs a more detailed description. How did you choose F_1,2 and why? Was the model trained by Hebbian learning?
    Can you show the concrete final model (its F-functions and weights)? Figure 3 is not comprehensible - could you describe more clearly what the four images represent?
2. There is no theoretical justification of the method (and its convergence) and the empirical evaluation is limited to a toy problem.
The paper addresses more limitations and problems of the proposed model than its advances.  Many important questions are not addressed:
 * Is it possible to extend the 2-bit parity model to the n-bit one? Are you able to show a solution for n=3?
 * The training process of the model is not described/analyzed.
 * The choice of F-functions for different tasks is not described/analyzed.
 * From the paper it seems like it is difficult to successfully  apply the model to different tasks. Is it right?


**Summary Of The Paper:**

Goals and contribution:
* The authors try to draw attention of the community to shallow recurrent neural networks trained by Hebbian learning in the context of cognitive material systems.
* They argue that these models could compete with deep-learning systems in the future  - thanks to upcoming advancement of in-materio technologies.
* They demonstrate this idea on an illustrative example of the N-bit (2-bit) parity function realized by the new-proposed SLIM (Shallow learning in Materio) model.
* The authors also point out several crucial open problems of the proposed approach as topics for future research.



**Summary Of The Review:**

Although the main idea of the paper is novel and interesting, its realization is poor. The proposed model is not well-described and it is not adequately empirically or theoretically justified.

---

> ### Author Response · Authors · 2022-11-19
> **Thank you**
>
> Your review was especially useful to help improve the manuscript. An updated version with a clearer Figure 3, a new Figure 4, some numerical justification is now provided. The link to Lawrence 2022b works now : https://openreview.net/forum?id=401LFvBGIb

---

### Official Review · Reviewer_h2U8 · 2022-10-23

**Confidence:** 4
**Correctness:** 3
**Technical Novelty And Significance:** 1
**Empirical Novelty And Significance:** 1
**Recommendation:** 1

**Clarity, Quality, Novelty And Reproducibility:**

The paper is written poorly and in a non-standard way, and cannot be accepted in its present form.
The work presented does not seem novel, or it is only marginally novel.
Almost no implementation details are given, and the only example is designed by hand.


**Strength And Weaknesses:**

Strengths:
- The paper is short, which makes it easy to read.

Weaknesses:
- The main problem tackled in the paper is not a particular concern since the early 80s and is largely solved.
- The structure of the paper is unorthodox and lacks most of the required content (concerning the theory and the model) and any practical evaluation.
- Figure 3 is difficult to understand, as no explanation is given about the notation in the text nor in the caption.
- The proposed architecture seems to be a standard RNN with a more flexible activation function, which however the authors leave undefined.
- Related literature is severely lacking, both to justify the significance of the problem addressed and all the work done on it since the 80s.


**Summary Of The Paper:**

The authors introduce a structural variant of single-layer perceptrons with added recurrent connections. The model seems to be mostly a standard RNN, with the different of a more general activation function.
No evaluation is provided, aside from a hand-designed example.

**Summary Of The Review:**

N/A

---

### Official Review · Reviewer_sByK · 2022-10-24

**Confidence:** 4
**Correctness:** 2
**Technical Novelty And Significance:** 1
**Empirical Novelty And Significance:** Not applicable
**Recommendation:** 1

**Clarity, Quality, Novelty And Reproducibility:**

The purpose of the paper is clear. Quality, novelty and reproducibility is hard to comment on since there is so little material.

**Strength And Weaknesses:**

Strengths:
The paper explores alternatives to the deep learning architecture and hopes to have self-organized nanomaterial-based learning machines.

Weaknesses:
The key idea is not so novel. From a multitude of recurrent neural networks (including the reservoir computing framework, mentioned by the authors) to predictive coding networks, many approaches utilize the idea of a network with internal dynamics, the equilibrium point of which helps perform the task.

There is very little work done to substantiate the claim for the particular architecture proposed.


**Summary Of The Paper:**

SLIM is intended to be a perceptron that involves a minimally connected recurrent network. The internal state variables have a nearest-neighbor interaction on a chain. The authors conjecture such networks could realize arbitrary N-bit Boolean functions, in contrast to the limitation of Rosenblatt perceptrons, as shown by Minsky and Papert.

**Summary Of The Review:**

This paper would need much more work to rise to the level of being worthy of consideration for publication at ICLR. At this point, it is just a scheme, and not a particularly novel one.

---

### Official Review · Reviewer_wULo · 2022-10-27

**Confidence:** 2
**Correctness:** 2
**Technical Novelty And Significance:** 1
**Empirical Novelty And Significance:** Not applicable
**Recommendation:** 3

**Clarity, Quality, Novelty And Reproducibility:**

Novel, but might profit from "a gifted mathematicians to make new connections or help neuroscientists in unravelling the mysteries of small-scale brains."


**Strength And Weaknesses:**

Strength:
* Out of the box

Weakness:
* Lack of box

**Summary Of The Paper:**

The paper suggests an alternative network architecture, but lacks analytical and numerical evaluation.

**Summary Of The Review:**

Ingenious, but in lack of a theory and of numerics

---

### Decision · Program_Chairs · 2023-01-20

**Decision:**

Reject

**Justification For Why Not Higher Score:**

This paper seems quite incomplete and lacking in technical detail. It’s presented in a confusing way and would benefit from more work and attention, and is thus not suitable for publication at this time.

**Justification For Why Not Lower Score:**

N/A

**Metareview: Summary, Strengths And Weaknesses:**

This is a rather unique paper, proposing a variant of single-layer perceptrons modified to include recurrent connections, as an alternative to standard deep learning architectures. It’s an interesting idea, but it seems undeveloped and vague. For instance, the original submission is only 3 pages of text.

All reviewers indicated that the paper is far too short and lacking in technical details. Having read the paper myself, I tend to agree. The figures especially are rather difficult to figure out, as they include only brief legends. The figures appear improved in the updated paper, but are still rather inscrutable. I agree with review joB1 that this paper is still a work in progress and would benefit from much more development, better presentation, and some sort of experimental evaluation of their proposed method. The reviewers, especially joB1, gave some great feedback and I urge authors to consider incorporating these in future submissions of this work.

Note that the authors added their rebuttals to the updated version of the paper itself, rather than via openreview, and hence the reviewers likely missed them initially.